# FAKE NEWS DETECTION VIA AN ADAPTIVE FEATURE MATCHING OPTIMIZATION FRAMEWORK

## ABSTRACT

The rampant proliferation of fake news across online platforms has become a significant cause for concern, necessitating the creation of robust detection techniques. Within the confines of this investigation, we present an optimization methodology built upon salient attributes tailored for the identification of fake news, spanning both unimodal and multimodal data sources. By harnessing the capabilities inherent in a diverse array of modalities, ranging from textual to visual elements, we are able to comprehensively apprehend the multifaceted nature of falsified news stories. Primarily, our methodology introduces an unprecedented array of features, encompassing word-level, sentence-level, and contextual features. This infusion bestows upon it a robust capacity to adeptly accommodate a wide spectrum of textual content. Subsequently, we integrate a feature-centric optimization technique grounded in the principles of simulated annealing. This approach enables us to ascertain the most optimal fusion of features, thereby mitigating potential conflicts and interferences arising from the coexistence of textual and visual components. Empirical insights garnered from exhaustive dataset experimentation decisively underscore the efficacy of our proposed methodology. Our approach outperforms standalone modalities as well as traditional single-classifier models, as evidenced by its superior detection capabilities. This research underscores the indispensable role played by the integration of multimodal data sources and the meticulous optimization of feature amalgamations. These factors collectively contribute to the creation of a resilient framework tailored for the identification of fake news within the intricate landscape of our contemporary, data-rich environment.

## 1 INTRODUCTION

In the contemporary landscape, where the Internet has firmly cemented its position as the dominant platform for social media interaction, employing the potential of Artificial Intelligence (AI) to combat the proliferation of fake information assumes paramount importance (Przybyla, 2020). Within this context, the integration of AI technologies into the domain of fake news detection emerges as a pivotal task. Recently, the pursuit of multi-modal fake news recognition introduces a plethora of intricate technological challenges. The integration of diverse data modalities, including text, images, and videos, necessitates the development of advanced algorithms capable of grasping the synergistic interplay among these disparate sources. Therefore, the main objective of this paper is to present intelligent mechanisms for fake news detection, a critical stride toward upholding the veracity and dependability of information.

In fact, assuring the precision of multi-modal fake news detection necessitates sophisticated techniques adept at managing the innate noise, ambiguity, and inconsistencies that frequently arise when dealing with multiple data types (Singhal et al., 2019). The coherent alignment of features extracted from diverse modalities, while accounting for potential disparities, presents a substantial computational hurdle. To confront these challenges, this paper focus on developing an Adaptive Feature Matching Optimization framework (AFMO) for fake news detection with both unimodal and multimodal resources. AFMO applies the potency of multiple modalities by employing textual and visual data to construct a holistic depiction of potential fake news. Specifically, distinct neural networks are leveraged to extract feature representations from the diverse modal information. Subsequently, an outlier detection algorithm is employed to eliminate training samples exhibiting

anomalous features, thereby enhancing the accuracy and dependability of the trained model. Additionally, a simulated annealing algorithm is employed to judiciously fuse features extracted from different modalities, thereby optimizing the overall model performance. Indeed, conventional fake news detection methodologies might confront visually salient image features that obscure crucial textual information, leading to an incomplete comprehension of the broader context. To address this obscurity, we adopt a feature-centric simulated annealing algorithm, aiming to ameliorate potential interference between textual and visual data through strategic feature selection. Consequently, the key of AFMO is to elevate the quality and discriminative prowess of the extracted features, all while alleviating the repercussions of cross-modal interference.

The structure of this paper is outlined as follows: Section 2 delves into the landscape of related research concerning fake news detection. In Section 3, an exposition of the proposed Adaptive Feature Matching Optimization (AFMO) framework is provided. Section 4 expounds upon the employed dataset, furnishing an analysis of baseline methodologies alongside the presentation of experimental outcomes. Section 5 concludes the paper's findings and draws its final insights.

# 2 RELATED WORK

## 2.1 FAKE NEWS DETECTION

The advent of deep learning heralded the introduction of Recurrent Neural Networks (RNNs) to uncover latent representations within textual features (Ma et al., 2016). Concurrently, some investigations incorporated Convolutional Neural Networks (CNN) for fake news detection by projecting each news event post onto a vector space and subsequently employing CNN to elicit text features from the resulting embedding matrix. These features were then channeled into a classifier for ultimate classification (Yu et al., 2017). An alternative strategy proposed a Graph Convolutional Network (GCN) model that conceptualizes news articles as graphs, with sentences serving as nodes and inter-sentence similarity as edges. This transformation reframed fake news detection as a graph classification predicament (Vaibhav et al., 2019). Furthermore, (Alzanin & Azmi, 2019) harnessed semi-supervised and unsupervised techniques to identify counterfeit news within social media.

In light of recent strides in deep learning methodologies, neural network models have ascended as the prevailing approach for fake news detection (Ma et al., 2019). Researchers have harnessed architectures such as CNN (Yu et al., 2017) and RNN (Ma et al., 2016) to dissect and unearth falsified information. Notwithstanding their promising outcomes, the lion's share of investigations has predominantly concentrated on textual attributes, often disregarding the latent advantages of incorporating image features (Su et al., 2019). In addition, some scholars use a generative approach (Yang et al., 2019) to do fake news detection. However, within the realm of social media, news articles often traverse broader spheres when accompanied by visual content (Ford et al., 2022), owing to their visual allure and the multifaceted ideas they encapsulate.

## 2.2 MULTIMODAL FAKE NEWS DETECTION

Nevertheless, the aforesaid methodologies predominantly apply single-modal data for fake news detection. Consequently, numerous researchers have shifted their focus towards the inclusion of images in counterfeit news detection, resulting in the proposition of multimodal detection frameworks (Qi et al., 2019). In the realm of multimodal fake news detection, systematic research experiments were conducted by Jin et al. (2017) to assess the role of images. Impressive results were achieved through an attention-based RNN multimodal fusion framework, effectively incorporating information from both textual and visual sources. A significant contribution, EANN, accentuates cross-domain fake news detection by eliminating domain-specific textual and visual information, while learning common feature representations across diverse domains (Wang et al., 2018). In a parallel vein, a 2019 study introduces a memory network module that engenders invariant features between different events for cross-domain recognition (Zhang et al., 2019). Diverging slightly, MVAE follows the blueprint of EANN, albeit its core architecture rests upon the foundation of Variational Autoencoder (VAE) utilizing word embedding vectors. It employs bidirectional LSTM to distill text and image representations from a pre-trained VGG-19 model. Subsequently, the concatenated hidden vector undergoes decoding to reconstruct the original samples. Further, these hidden vectors traverse two fully connected layers to facilitate fake news detection (Khattar et al., 2019).

Regarding the amalgamation of diverse modalities as potentially disruptive noise, CARMN capitalizes on both the Cross-Modal Attention Residual Network (CARN) and the Multichannel Convolutional Neural Network (MCN). CARN adeptly integrates pertinent information across modalities while preserving the distinctive attributes of each. Meanwhile, MCN extracts feature representations from both the initial and fused textual data (Song et al., 2021). Another approach, presented in Wu et al. (2021); Qian et al. (2021), is MCAN. It adopts the Co-Attention (CA) block as the foundational unit of their Multi-modal Co-Attention Network. The CA block encompasses two parallel CA modules—one catering to the textual domain and the other to the visual domain—processing textual and visual information respectively. Within the CA block, one domain's features are treated as query features. By continually intertwining image and text data, the approach emulates the manner in which humans consume news, harmonizing both imagery and text, thereby elevating the performance of fake news detection (Wu et al., 2021). Furthermore, a two-fold inconsistency detection strategy was proposed by Xiong et al. (2023) to curate noise stemming from the feature fusion process.

## 2.3 Simulated Annealing

The simulated annealing algorithm stands as a versatile stochastic search technique extensively employed across a spectrum of combinatorial optimization problems. Its adaptability renders it a valuable asset in various domains, including VLSI design, image recognition, and research within the realm of neural network computing. This algorithm has found prominent application in tasks such as clustering (Lee & Perkins, 2021), the most influential node in the network (Jiang et al., 2011). Additionally, researchers have amalgamated simulated annealing with other methodologies to attain heightened performance vis-à-vis singular approaches, particularly in specified domains. For instance, the convergence of simulated annealing with genetic algorithms has been harnessed to solve the multi-class multidimensional knapsack optimization problem (Meng et al., 2019). Furthermore, the amalgamation of simulated annealing with the Salp Swarm Algorithm (SSA) and genetic algorithm has been wielded to calibrate the equilibrium between exploration and exploitation within SSA (Kassaymeh et al., 2022). Buoyed by the demonstrated efficacy of the annealing algorithm within the ambit of combinatorial optimization, we judiciously integrate this algorithm into the proposed AFMO framework, thereby surmounting the challenge of synergistically assimilating textual and visual features.

## 3 Methodology

### 3.1 Overview of AFMO

Consider a news dataset $D$ comprising $N$ samples, each consisting of a news text, news image, and corresponding true or false label. The proposed AFMO primarily applies the textual and image information of the news to infer its label. The principal formulas employed in this article are delineated in Table 1.

| Variables | Explanation |
|---|---|
| $D$ | News dataset with N elements. |
| $txt_i$ | The text of the i-th news item. |
| $img_i$ | The image of the i-th news item. |
| $label_i$ | The label of the i-th news. |
| $m^t$ | Text features of all news. |
| $m^p$ | Image features of all news. |
| $D_{reduce}$ | Training set after removing outliers. |
| $m_i^t$ | Value of the j-th dimension in the text feature. |
| $m_i^p$ | Value of the j-th dimension in the image feature. |
| $x$ | Indicates which text features were selected. |
| $y$ | Indicates which image features were selected. |
| $m_i^r$ | Selected features after processing. |

Table 1: Description of mathematical notations.

As illustrated in Figure 3, the news articles undergo a process of feature extraction utilizing the BERT and VGG modules, which are responsible for extracting textual and image information, re-

spectively. These modules generate comprehensive representations of the respective text and image modalities. Following this, the outlier removal module is engaged to identify and eliminate instances featuring anomalous attributes, thereby yielding a novel training set that undergoes subsequent processing. Ultimately, the simulated annealing algorithm is employed to judiciously select and fuse pertinent information emanating from diverse facets of the text and image modalities. The amalgamated features are subsequently funneled into a classifier to yield the ultimate predictions. In light of the potential existence of data samples exhibiting atypical characteristics within the original dataset, we apply the entirety of text and image features within the training set as input. Employing methodologies such as the Mahalanobis distance or the K-Nearest Neighbors (KNN), we discern and expunge these aberrant instances, resulting in a refined dataset denoted as $D_{reduce}$. Subsequently, the training regimen is executed utilizing this curated dataset. The simulated annealing algorithm assumes a pivotal role in orchestrating the fusion of text features $m^t$ and image features $m^p$, thereby orchestrating the selection of pertinent attributes. Conclusively, the amalgamated features traverse an array of fully connected layers to culminate in the ultimate predicted outcomes.

## 3.2 FEATURE EXTRACTION

### 3.2.1 TEXT FEATURE

BERT (Bidirectional Encoder Representations from Transformers) (Devlin et al., 2019) stands as a transformative model grounded in the principles of attention mechanisms, and it is pre-trained via the Masked Language Model (MLM) task. In our investigation, we designate $txt_i$ as the input for BERT, given that the terminal layers of BERT encapsulate a comprehensive textual mosaic. Empirical findings have consistently demonstrated that the outcomes of the last quartet of hidden layers offer optimal performance across diverse NLP tasks, in contrast to the results generated by other layers. Accordingly, we concatenate the outputs from these final four hidden layers, thereby cultivating elemental textual features, resulting in a cumulative tally of four outputs. These outputs are sequentially concatenated to form the complete basic text feature, denoted as $m_i^t$, as shown in the following equation:

$$m_i^t = TokenEmbeddings(txt_i) + PositionEmbeddings(txt_i) + SegmentEmbeddings(txt_i).$$

Here, $m_i^t$ represents the embedding vector of the $i$-th text sample, $TokenEmbeddings$ corresponds to the initial embedding vector of the token, $PositionEmbeddings$ represents the position embedding vector of the token, and $SegmentEmbeddings$ indicates the segment embedding vector of the token.

### 3.2.2 IMAGE FEATURE

VGG Simonyan & Zisserman (2015), acclaimed for its profound network architecture, attains depth through the arrangement of multiple VGG blocks, each comprising convolutional and pooling layers. This design imparts robust generalization capabilities, rendering it efficacious across a diverse array of image datasets. Thus, VGG is conventionally harnessed for the extraction of image features. Within our study, we employ VGG-19, pre-trained on the ImageNet dataset, as the designated image information feature extractor. When presented with an input image of dimensions $W \times H$, a sequence of convolutional and pooling layers collaborates to engender a feature vector of dimension $C$, encapsulating the salient image characteristics. In this context, $C$ corresponds to the channel count within the ultimate convolutional layer. Within our methodology, $img_i$ assumes the role of input for VGG-19. Aligned with the structural blueprint of the VGG network, the terminal layer's output encapsulates an all-encompassing spectrum of image traits. As a result, we adopt the terminal layer's output from VGG-19 as our image features, a representation realized through the following equation:

$$m_i^p = VGG19\,(img_i)\,.$$

## 3.3 ENHANCEMENT OF TRAINING

The text features $m_i^t$ and image features $m_i^p$ of each sample are concatenated to form $m_i$, where $m_i = m_i^t \bigoplus m_i^p$. Consequently, all the features of the samples are represented as $M = \{m_i \mid \forall i = 1, \ldots, N\}$.

The Mahalanobis distance serves as a conventional metric to gauge the separation between a specific point and a distribution. In our methodology, we apply the Mahalanobis distance computation to discern and subsequently eliminate anomalous data instances. Indeed, we embark by deriving the mean vector $\mu$ and the covariance matrix $\Sigma$ of the feature set $M$. Thereafter, the distance between each sample and the dataset as an aggregate entity is meticulously computed. The threshold, denoted as $Dist_{mean}$, is ascertained by calculating the average distance across all samples and then multiplying it by the standard deviation $\sigma$, adjusted by a weighting factor $\alpha$. The default value for the weighting coefficient $\alpha$ is set at 1. Instances where the distance between a sample point and the dataset surpasses this threshold are deemed abnormal and are subsequently excluded from consideration.

The Mahalanobis distance $Dist(m_i)$ is computed as follows:

$$Dist\left(m_i\right) = \sqrt{\left(m_i - \mu\right)^T \Sigma^{-1} \left(m_i - \mu\right)}.$$

The threshold for abnormality, $Dist_{max}$, is obtained by adding the product of $\alpha$ and $\sigma$ to $Dist_{mean}$:

$$Dist_{max} = Dist_{mean} + \alpha \cdot \sigma.$$

KNN (k-nearest neighbors) emerges as a prevalent unsupervised clustering algorithm, seamlessly extending its utility to encompass outlier detection. The foundational tenet revolves around gauging the distance between the present sample and the entirety of the dataset. Divergent from quantifying distances between points and distributions, KNN computes distances between each individual sample and the remaining samples, employing the Euclidean distance formula, illustrated through the ensuing equation:

$$dist\left(m_i, m_j\right) = \left(\sum_{l=1}^{n} \left(m_i^l - m_j^l\right)^2\right)^{\frac{1}{2}}.$$

The $k$ samples characterized by the closest distances are meticulously chosen, and the arithmetic mean of the distances between the current sample and these $k$ samples is meticulously computed. These average distances are then amassed and ordered in descending order, ranging from the largest to the smallest values. By delineating a threshold to demarcate the proportion of outliers, we proceed to ascertain whether the average distance associated with the current sample surpasses this predefined threshold. Should the average distance exceed the set threshold, the current sample is categorically categorized as an outlier. During the ultimate phase, these outlier samples are systematically excised, culminating in the inception of a fresh dataset termed $D_{reduce}$. This distilled dataset subsequently forms the foundation for subsequent training and testing phases, employing the identical test set utilized within the original dataset.

### 3.4 METAHEURISTIC FOR FEATURE SELECTION

Within the original dataset, each text feature $m_i^t$ is epitomized by a collection of values $\left\{f_1^t, f_2^t, \ldots, f_{n_{txt}}^t\right\}$, wherein $n_{txt}$ denotes the extent of the text feature. In parallel, every image feature $m_i^p$ is characterized by a set of values $\left\{f_1^p, f_2^p, \ldots, f_{n_{img}}^p\right\}$, with $n_{img}$ reflecting the dimensionality of the image feature. Traditional feature selection tactics encompass the aggregation of the two modality features, a practice that may inadvertently culminate in the loss of information, particularly when the feature vectors embody divergent connotations. Nonetheless, this approach exponentially escalates computational complexity and may inadvertently incorporate irrelevant or spurious information from all dimensions, thereby potentially compromising outcomes.

To address this quandary, we advocate the adoption of the simulated annealing algorithm for feature selection, poised to sieve out efficacious dimensions whilst discarding those devoid of relevance. The simulated annealing algorithm exhibits the capacity to transcend local optima, thereby accommodating suboptimal solutions during the exploration process and evading the entrapment within local optima. Through this approach, we aspire to elevate the caliber of selected features, curtail possible interferences between textual and visual insights, and augment the overall efficacy of classification tasks. The principal strides of the simulated annealing algorithm for feature matching are succinctly encapsulated as follows.

**Heating Up Process**: Initially, an initial temperature $t_0$, a minimal temperature $t_{min}$, and a prevailing temperature $t_{cur}$ (initially aligning with $t_0$) are predetermined. For every combination of

$m^t \bigoplus m^p$, a binary sequence $x \bigoplus y$ is conceived, the length of which corresponds to the collective dimensionality of the encompassed features. Within this sequence, $x_i$ assumes the value 1 if the $i$-th dimension of the text feature is adjudged pertinent for the ultimate rumor detection, retaining its inherent value. In contrast, $x_i$ adopts the value 0 if the $i$-th dimension lacks significance, compelling its value to be set at 0. A completely arbitrary $x \bigoplus y$ configuration is initialized, serving as the foundational state.

**Isothermal Process**: If the current temperature $t_{cur}$ is less than the minimum temperature $t_{min}$, the iteration process draws to a close. Conversely, if this condition remains unmet, a fresh $x \bigoplus y$ configuration is engendered grounded in its predecessor. This metamorphosis is effectuated by electing a certain number of dimensions from the initial $x \bigoplus y$ arrangement and subsequently inverting their prevailing 0-1 values. The extent of randomness and the tally of selected dimensions escalates in tandem with elevated temperatures. This escalating relationship is mathematically formulated by the ensuing equation:

$$length_{change} = \frac{t_{cur} - t_{\min}}{t_0 - t_{\min}} \times (N_{txt} + N_{img}).$$ (1)

The updated $x \bigoplus y$ configuration is employed to transform the initial features $m_i$, yielding the reconfigured features $m_i^r$. Subsequently, these reconfigured features are introduced into the classifier detector, leading to the derivation of the classification outcome $out_i$. The accuracy rate is meticulously computed through the juxtaposition of $out_i$ against the actual label $label_i$. This accuracy rate emerges as the pivotal objective function governing the simulated annealing algorithm:

$$out_i = detector\left(m_i^r\right),$$ (2)

where,

$$m_i^r = \left(x \cdot m^t, y \cdot m^p\right).$$ (3)

**Cooling Process**: If the accuracy rate of the current iteration is higher than the previous optimal accuracy, the revised $x \bigoplus y$ configuration is directly embraced. In contrast, if the accuracy rate falls below, the determination to embrace the fresh solution transpires probabilistically, hinging on the prevailing temperature $t_{cur}$ and the discrepancy in accuracy between the existing and preceding optimal accuracy, represented by $\Delta E$. Elevated temperatures engender heightened probabilities of adoption, effectively fostering the avoidance of local optima in favour of global optima. The probability of integrating the prevailing solution is governed by the ensuing equation:

$$P(x' \bigoplus y' \to x \bigoplus y) = e^{-k \cdot \frac{\Delta E}{t_{cur}}}.$$ (4)

If the current solution is accepted, its accuracy rate assumes the mantle of the optimal accuracy rate. Subsequently, the prevailing temperature is multiplied by the cooling factor denoted as $k$, and this updated temperature is funneled back into the Isothermal Process phase. In the event that the current solution is not assimilated, the progression reverts promptly to the Isothermal Process stage.

The above three steps are repeated multiple times to complete the multi-round simulated annealing algorithm, which aims to achieve the best results.

## 4 EXPERIMENTAL RESULTS

### 4.1 DESCRIPTION OF DATASET

In the context of unimodal testing, we opted to assess the efficacy of the proposed Adaptive Feature Matching Optimization (AFMO) framework using the PolitiFact dataset (Shu et al., 2020). Acquired from FakeNewsNet, the PolitiFact dataset encompasses news articles drawn from the fact-checking website PolitiFact, meticulously labeled for their authenticity by domain experts. For the multimodal scenario, we selected two extensively recognized and publicly accessible benchmark datasets—namely, the Chinese Weibo dataset and the English Gossipcop dataset—for our rigorous evaluation and experimentation in the domain of fake news detection. To offer a concise overview of the dataset characteristics, we present a summary of the dataset statistics in Table 2.

| Dataset | Fake News | Real News | Texts | Images |
|---|---|---|---|---|
| PolitiFact | 463 | 373 | 836 | - |
| Chinese Weibo | 4108 | 3615 | 7723 | 7723 |
| English Gossipcop | 3398 | 12365 | 15763 | 15763 |

Table 2: The statistics of considered datasets.

## 4.2 EXPERIMENT SETUP

The division of the dataset into training, validation, and test sets follows a ratio of 7:1:2, ensuring an appropriate distribution. The computational resources harnessed for our experimentation encompass an Intel(R) Xeon(R) Gold 6248R CPU with a clock speed of 3.00GHz, coupled with an NVIDIA A100 80GB PCIe GPU.

Concerning the text modality, we leverage the sentence-level vector outputs from the last four layers of the BERT model to serve as the textual feature representation. These representations undergo processing through a tandem of fully connected layers, culminating in the derivation of text features with a dimensionality of 64. Concurrently, for the image modality, the terminal layer's output within the FEATURES layer of the VGG19 model assumes the role of image feature representation. Analogous to the text features, a solitary fully connected layer facilitates the creation of image features with a dimensionality of 64. To counteract potential overfitting, both the BERT and VGG models incorporate parameter freezing. In addition, a dropout layer with a dropout rate of 0.5 is strategically interposed following each fully connected layer. The chosen batch size is 90, while the optimization scheme entails Adam with an initial learning rate established at 0.001. The training trajectory encompasses 100 epochs.

Among them, the selection of the KNN parameter K value and the weight value of the standard deviation of the Mahalanobis distance used in the Reliability Enhancement module will affect the final results of the experiment. Considering that the method of using bayes tuning parameter will run the whole process extremely time-consuming every time after adjusting the parameters, this experiment manually selects the KNN parameter K value and the value of the standard deviaation weight of the Mahalanobis distance. As shown in Figure 1 and Figure 2, it demonstrates the model prediction effect of K from 1 to 10 and the value of Mahalanobis distance standard deviation weights $\alpha$ from 0.1 to 1.5 on gossipcop dataset. Based on the results, the KNN parameter K = 3 and the Mahalanobis distance standard deviation weight value alpha = 0.8 were chosen.

For the integrated simulated annealing algorithm, the initial temperature is set to exp(4), the minimum temperature is defined as exp(-1), and the temperature decay coefficient is stipulated at 0.98. Given the classification nature of our experiment, the evaluation metrics are aptly chosen to encompass Accuracy, Precision, Recall, and F1 score. Furthermore, the versions of the pivotal Python packages engaged in this research is provided as follows: python=3.7.4, pytorch=1.11.0, cuda=10.2, torchvision=0.11.2+cu102.

## 4.3 BASELINE MODELS

We compare our approach with the state-of-the-art methods and some baselines, as listed below:

- for unimodal, **Textual** (Kim, 2014), **Visual** (Simonyan & Zisserman, 2015), **Text-RF** (Shrestha & Spezzano, 2021), **LR-Bias** (Shrestha et al., 2020), **XGBoost** (Shrestha & Spezzano, 2021), **LSTM-ATT** (Shrestha & Spezzano, 2021), **GRU-2** (Ma et al., 2016), **GCAN** (Lu & Li, 2020);

- for multimodal, **VQA** (Antol et al., 2015), **ATT-RNN** (Jin et al., 2017), **EANN** (Wang et al., 2018), **MVAE** (Khattar et al., 2019), **MKN** (Zhang et al., 2019), **CARMN** (Song et al., 2021), **TRIMOON** (Xiong et al., 2023).

## 4.4 RESULT AND ANALYSIS

### 4.4.1 ABLATION EXPERIMENTS ANALYSIS

In this investigation, we adopt a comprehensive set of evaluation metrics encompassing Accuracy, Precision, Recall, and F1 score to comprehensively assess the efficacy of the model. Figure 4 depicts the evolution of selected evaluation metrics with increasing iterations during the execution of the simulated annealing algorithm. The figure prominently elucidates how the simulated annealing algorithm dynamically refines accuracy, precision, recall, and F1 score, progressively enhancing the outcomes in comparison to the initial stage.

### 4.4.2 CASE STUDY

For the case of instance "Id: 1241528475" from the English Gossipcop dataset, the original model exhibited a failure in detecting the fake news example "Please note that this form cannot be used to reset your Googleor Facebook password. Visit Google or Facebook to do that." However, conventional fake news detection methods deemed the news as authentic, though it was, in fact, a fake news. Notably, our proposed model effectively identified this news as fake. The textual content of the story indicated that the form displayed in the image could not be used for password reset and required a website visit for the same. Intriguingly, the image portrayed an overweight individual holding a child, which was unrelated to the textual content. After the incorporation of the simulated annealing algorithm, the fake news was aptly detected. This scenario highlighted how the presence of a seemingly normal image accompanying the fake news had interfered with the detection process. The utilization of the simulated annealing algorithm in our approach successfully mitigated this interference, resulting in improved detection accuracy.

### 4.4.3 COMPARATIVE ANALYSIS

A comprehensive performance comparison and analysis of our approach against various baseline models was undertaken. The data for these baseline models were sourced from their respective papers, and the results are detailed in Table 3. The outcomes are strikingly illustrative of the robust performance of our solution across both datasets. On the PolitiFact dataset, our model excels in comparison to other models employing traditional machine learning and deep learning techniques, as indicated by its superior performance across all four evaluation metrics. Notably, it attains a substantial accuracy enhancement of $8.47\%$ when compared to the leading XGBoost models. Furthermore, in terms of recall rate and F1 score, our model outperforms the competitive GCAN model by $4.62\%$ and $6.5\%$ respectively. These findings strongly affirm the efficacy and credibility of our devised fake news detection model.

| Model | Accuracy | Precision | Recall | F1-score |
|-------|----------|-----------|--------|----------|
| Text-RF | 0.814 | 0.803 | 0.773 | 0.787 |
| XGBoost | 0.832 | 0.836 | 0.832 | 0.829 |
| LSTM-ATT | 0.820 | 0.835 | 0.820 | 0.816 |
| GRU-2 | 0.749 | 0.709 | 0.705 | 0.704 |
| GCAN | 0.808 | 0.795 | 0.841 | 0.835 |
| AFMO | **0.917** | **0.913** | **0.887** | **0.900** |

Table 3: Performance of baseline models and AFMO on the PolitiFact dataset.

In order to ascertain the efficacy of AFMO across both English and Chinese datasets, we carried out experiments on both and juxtaposed the results with baseline models. As demonstrated in Table 4, when considering the Weibo dataset, models reliant on a singular modality typically exhibit lackluster performance, particularly those hinging solely on image information, achieving a mere 59.4% accuracy. This observation indicates that prevailing models inadequately capitalize on the insights offered by images. The erstwhile TRIMOON model had established itself as the pinnacle with a 91.26% accuracy on this dataset, holding an advantage of at least 4% over other methodologies. By contrast, AFMO surpasses all contenders, registering the highest accuracy at 92.2%. Furthermore, AFMO attains commendable precision, recall, and F1 scores of 93.7%, 91%, and 92.4%, respectively. These findings underscore the superiority of AFMO relative to existing models.

| Methods | Accuracy | Precision | Recall | F1-score |
|---------|----------|-----------|--------|----------|
| Textual | 0.764 | 0.776 | 0.721 | 0.747 |
| Vis | 0.594 | 0.583 | 0.752 | 0.657 |
| VQA | 0.579 | 0.581 | 0.665 | 0.620 |
| ATT-RNN | 0.784 | 0.797 | 0.781 | 0.789 |
| EANN | 0.807 | 0.831 | 0.788 | 0.809 |
| MVAE | 0.681 | 0.756 | 0.589 | 0.662 |
| MKN | 0.792 | 0.805 | 0.788 | 0.796 |
| CARMN | 0.869 | 0.891 | 0.814 | 0.851 |
| TRIMOON | 0.913 | 0.930 | 0.888 | 0.909 |
| AFMO | **0.922** | **0.937** | **0.910** | **0.924** |

Table 4: Performance of baseline models and AFMO on the Weibo Dataset.

Likewise, with respect to the Gossipcop dataset as delineated in Table 5, AFMO exhibits the highest accuracy, and outperforms other models in terms of recall and F1 score, amassing an impressive 97.1% and 92.6% respectively. Additionally, AFMO achieves accuracy and precision rates of 87.5% and 88.5% correspondingly. Although AFMO falls behind TRIMOON in precision, it surmounts all models, including TRIMOON, in terms of accuracy, recall, and F1 score. Notably, AFMO boasts one of the most substantial recall performances, outpacing all other models by approximately 10%.

| Methods | Accuracy | Precision | Recall | F1-score |
|---------|----------|-----------|--------|----------|
| Textual | 0.838 | 0.966 | 0.847 | 0.903 |
| Vis | 0.779 | 0.878 | 0.843 | 0.860 |
| VQA | 0.779 | 0.873 | 0.847 | 0.860 |
| ATT-RNN | 0.825 | 0.914 | 0.868 | 0.890 |
| EANN | 0.796 | 0.877 | 0.862 | 0.870 |
| MVAE | 0.822 | 0.919 | 0.861 | 0.889 |
| CARMN | 0.851 | 0.942 | 0.875 | 0.907 |
| TRIMOON | 0.869 | 0.963 | 0.880 | 0.907 |
| AFMO | **0.875** | 0.885 | **0.971** | **0.926** |

Table 5: Performance of baseline models and AFMO on the Gossipcop Dataset.

## 5 CONCLUSION

In this research, we introduce a novel approach for training models to detect fake news and for strategically combining diverse modalities to identify misleading information. Our method comprises several key steps. Initially, we extract essential text and image features from news articles by applying BERT and VGG19, respectively. Following this, we employ robust outlier detection techniques grounded in KNN and Mahalanobis distance to effectively eliminate training samples exhibiting abnormal features. The core of our framework lies in the utilization of the simulated annealing algorithm, which plays a crucial role in selecting the most informative features that effectively combine text and image modalities. The overarching objective here is to minimize potential interference between textual and visual information, thereby elevating the overall quality and discriminability of the selected features. These meticulously chosen features are then seamlessly integrated into the classification process for the purpose of identifying false news. Our approach is substantiated through rigorous experimentation, wherein we compare its performance to existing methodologies. The empirical results strongly underline the effectiveness and superiority of our approach over alternative methods in the domain of fake news detection. As we look ahead, our research trajectory encompasses the integration of user-related attributes, including metrics such as follower and friend counts. By incorporating this additional layer of user-centric information, we envision a further augmentation of accuracy and overall performance in the realm of false news detection. This holistic approach, encompassing diverse modalities and user attributes, holds the promise of yielding even more impressive results in addressing the multifaceted challenge of identifying and mitigating fake news.

ACKNOWLEDGMENTS

We would like to thank the anonymous reviewers for their relevant and rich remarks that allowed us to improve the presentation of our results.

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

## A  APPENDIX

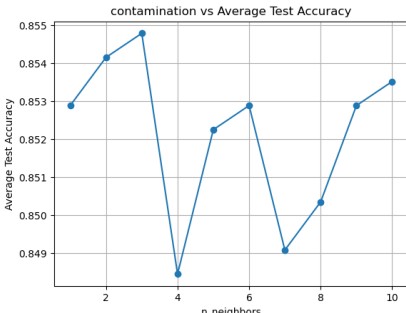
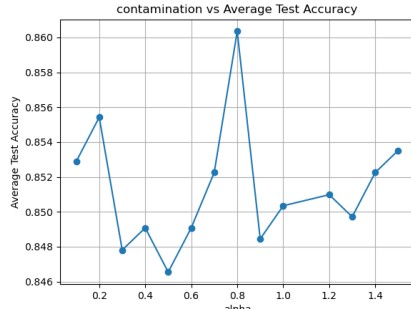

Figure 1: the model prediction effect of K from 1 to 10

Figure 2: the model prediction effect of $\alpha$ from 0.1 to 1.5

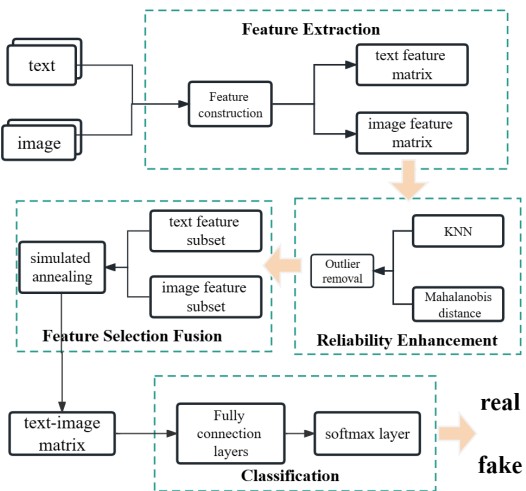

Figure 3: Overview of AFMO.

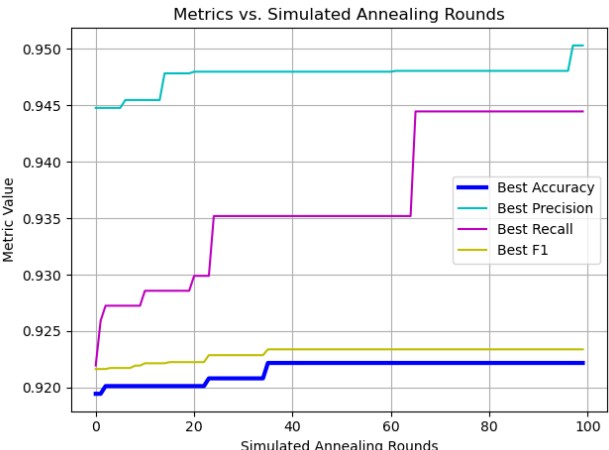

Figure 4: Changes in selected evaluation metrics as the iteration of the simulated annealing algorithm increases.

