# OpenReview forum: "Fake News Detection via an Adaptive Feature Matching Optimization Framework"
_ICLR.cc/2024/Conference — ICLR 2024 Conference Withdrawn Submission_

### Official Review · Reviewer_PUtn · 2023-10-26

**Soundness:** 1 poor
**Presentation:** 2 fair
**Contribution:** 1 poor
**Rating:** 3
**Confidence:** 4

**Summary:**

Inspired by the fake news detection task, the paper investigates the use of simulated annealing to perform selection of text and image features while training a multimodal neural network. The proposed framework, Adaptive Feature Matching Optimization (AFMO), consists of extracting text and image features using well-known pretrained models (BERT and VGG), removing sample outliers, and then training a feed-forward neural network while the set of active features in the input is modified by an algorithm whose proposals are accepted or rejected through simulated annealing (SA) based on the loss. The fraction of features which are "flipped" in the proposal decreases from 1.0 to 0.0 linearly as the temperature goes from t0 to tmin. The authors evaluate AFMO performance for text-only data on PolitiFact and for image+text data on Gossipcop and one of the Weibo datasets. The results indicate that AFMO outperforms all the baselines w.r.t. Accuracy, Recall and F1-score.

**Strengths:**

S1. While simulated annealing for feature selection has been explored by other works, this idea doesn't seem to be too explored in the context of neural networks with noisy inputs (perhaps due to the challenge of stabilizing the optimization).

**Weaknesses:**

W1. The text does not provide enough details for implementing the proposed feature selection algorithm.

W2. Some methodological issues, particularly regarding the use of datasets.

W3. No ablation study to understand the impact of feature selection.

W4. Novelty seems to be limited to decreasing the number of flipped features during annealing.

W5. Text is imprecise at some points and needlessly elaborate overall.

W6. Paper does not provide a way to generalize to datasets that contain environment features (e.g., comments, likes, etc).

**Questions:**

Q1. Many details have been left out.
- For text features, do you take the embeddings corresponding to all positions (instead of the usual approach of taking just the first position) and, if so, why?
- Why do you need segment embeddings if the task doesn't seem to involve multiple text segments in each observation?
- Can you describe the "array of fully connected layers" at the end of the each feature extractor and that at the end of classifier?
- Does the temperature change at the end of an iteration (minibatch) or at the end of an epoch?

Q2. Please address the following concerns:
- For KNN, you don't need to normalize the dimensions by variance?
- Which multimodal Weibo dataset did you use? Jin et al., 2017 or Zhang et al., 2021a?
- Why the statistics in Table 2 do not match those in other papers (see Shu et al. 2018)? Is it showing the reduced data? Could this also explain the discrepancies between the results in Tables 4-5 and those in Hu et al., Deep learning for fake news detection: A comprehensive survey, AI Open, 2022?
- Does AFMO reduce to a BERT classifier when there is only textual data? If so, what could explain the fact that such simple model it is outperforming all the baselines on PolitiFact by a wide margin; don't you need to include a stronger baseline (e.g., dEFEND)? Or does AFMO still includes the feature selection step?

Q3. What is the performance gain of the feature selection step? What are potential alternatives for feature selection?

Q4. Please clarify what exactly are the novel contributions of the paper.

**Details Of Ethics Concerns:**

No ethical concerns.

---

> ### Author Response · Authors · 2023-11-23
>
> In the new version, we have updated the literature review and experimental sections.
>
> Comment 1.1: For text features, the mean of embeddings from all segments of the text is used to represent the overall embedding of that text.
>
> Comment 1.2: "Array of fully connected layers" refers to using multiple layers of fully connected layers to perform the final classification on the processed features.
>
> Comment 1.3: The temperature changes in every iteration, i.e., after the accuracy is optimized or with a certain probability each time.
>
> Comment 2.1: KNN was used without normalization of dimensions because the same method was used to extract features.
>
> Comment 2.2: The Weibo dataset used is from Jin et al., 2017.
>
> Comment 2.3: The statistical data in Table 2, except for Gossipcop, was obtained from open-source datasets by other authors. Gossipcop was crawled based on FakeNewsNet mentioned in the paper. FakeNewsNet is a web crawler on websites such as Twitter. The crawled content may be missing text or images due to reasons such as the news being banned by Twitter or the original post being deleted by the user or the image being expired. Therefore, after crawling, news data with empty text or images were removed, which may cause some differences in statistics from the dataset.
>
> Comment 2.4: AFMO also includes steps of abnormal feature elimination and retraining, as well as annealing feature fusion when there is only text data.
>
> Comment 3: The performance gain of feature selection is around 2%.
>
> Comment 4: Outlier detection algorithms are used to eliminate training samples that exhibit abnormal features, thereby improving the accuracy and reliability of the trained model. In addition, simulated annealing algorithms are employed to wisely fuse features extracted from different modalities, filtering out image features that may visually stand out and affect key textual information, thus optimizing the accuracy and reliability of the model.

---

### Official Review · Reviewer_eXUi · 2023-10-30

**Soundness:** 2 fair
**Presentation:** 2 fair
**Contribution:** 2 fair
**Rating:** 3
**Confidence:** 4

**Summary:**

The author has introduced an optimization method, the core of which lies in the utilization of the simulated annealing algorithm. The objective is to filter out the most informative  features of text and image features and reduce potential interference between text and visual
information.

**Strengths:**

1. The paper is generally structured clearly.
2. Experimental results show that the proposed model has comparable or improved performance.

**Weaknesses:**

1. The advantages and innovativeness of the proposed method appear to be somewhat limited.
2. The placement of relevant figures in the appendix makes for slightly inconvenient reading, particularly with respect to Figure 1.
3. In the experimental section, some of the recent models proposed in 2022 and 2023 have not been compared, especially the unimodal methods.
4. The position of comparative analysis（4.4.3） seems unreasonable and should be mentioned in 4.4.1.

**Questions:**

1. The proposed model seeks to minimize potential interference between textual and visual information, but whether the feature selection through simulated annealing algorithm actually achieves the above-mentioned purpose seems to lack a detailed mechanism explanation.
2. What are the advantages compared to using neural networks like attention mechanisms to fuse features?
3. Which dataset was the experiment in Figure 2 done on? It should be noted in the text that it would be better to perform the same ablation experiments on the other two datasets.
4. The example in the 4.4.2 case study seems to be able to identify correctly using some methods based on capturing the similarity or ambiguity of images and texts(e.g. Cross-modal Ambiguity Learning for Multimodal Fake News Detection). I don’t know if there is a similar model in the baseline selected by the authors. This example does not seem to explain the advantages of the simulated annealing algorithm very well.
5. Are abnormal data removed during the running process of the comparison algorithms in Tables 4 and 5?

---

> ### Author Response · Authors · 2023-11-23
>
> In the new version, we have updated the literature review and experimental sections.
>
> Comment 1: The use of annealing algorithms can eliminate the mutual interference generated during multimodal feature fusion.
>
> Comment 2: Using accuracy as the objective function in simulated annealing is fair to other baselines, as it represents the main direction of optimization. AFMO has higher accuracy than almost all other baselines, and the comparison with other baselines in this article is mainly based on the value of accuracy.
>
> Comment 3: The ablation experiment was conducted on the weibo dataset.
>
> Comment 4: The baseline does not include a model for capturing image and text similarity.
>
> Comment 5: Yes, we removed the abnormal features.

---

### Official Review · Reviewer_Snvc · 2023-10-31

**Soundness:** 2 fair
**Presentation:** 2 fair
**Contribution:** 2 fair
**Rating:** 3
**Confidence:** 5

**Summary:**

The paper presents a comprehensive optimization methodology specifically designed for fake news detection, capable of handling both unimodal and multimodal data sources. The framework is structured in four sequential steps: feature extraction, outlier removal, feature fusion, and classification. The paper validates its approach by conducting experiments on three diverse datasets: PolitiFact, Weibo, and Gossipcop. The empirical results show that the proposed framework consistently outperforms existing baselines in key metrics such as accuracy, precision, recall, and F1 score. Overall, the paper makes a robust contribution to the area of fake news detection by introducing a multi-faceted, effective methodology.

**Strengths:**

1. The paper leverages a variety of text features, including word-level, sentence-level, and contextual features, which help to understand the text content.
2. The paper employs a simulated annealing algorithm to optimize the feature fusion process, which is a novel and effective technique for this task.
3. The paper conducts extensive experiments on three datasets with different languages and domains, and demonstrates the superiority of the proposed framework over existing methods.

**Weaknesses:**

1. The paper does not provide enough details about the outlier removal step, such as how to choose the threshold for Mahalanobis distance and the number of neighbors for KNN.
2. The baseline compared in this paper is relatively weak. As far as I know, there are many more advanced multi-modal fake news detection works.
3. The paper does not provide any qualitative examples or visualizations to illustrate how the framework works and why it is effective.

**Questions:**

1. What do you mean by "aberrant instances", is it deleting part of the dataset or part of the features?
2. What are the shortcomings and advantages of your method compared with other methods for eliminating multi-modal feature interference?
3. Why are the parameters of simulated annealing chosen in this way? Is there a better option? Will the cost be too high?

---

> ### Author Response · Authors · 2023-11-23
>
> In the new version, we have updated the literature review and experimental sections.
>
> Comment 1: Provide the threshold value of Mahalanobis distance (0.08) and the number of domains for KNN (domain number k=3). Currently, the threshold value is set manually, such as 0.08 in the experiment. Different threshold values will result in different abnormal features being removed, which will affect the performance of the final retrained model. Bayesian parameter tuning can be used to find the optimal threshold ratio, but the time cost will increase exponentially.
>
> Comment 2: Baseline selected the most advanced CARMN (2021) and TRIMOON (2023) in recent years. In the TRIMOON paper, it was mentioned that it was already the most advanced model, so no new baselines were added.
>
> Comment 3: An exception instance refers to some characteristics.
>
> Comment 4: Annealing has advantages and disadvantages in eliminating multimodal feature interference: it can accurately capture and eliminate interference, but it takes a long time.
>
> Comment 5: Using Bayesian parameter tuning to find the optimal parameters can be very time-consuming.

---

> > ### Comment · Reviewer_Snvc · 2023-11-23
> >
> > Dear authors,
> >
> > Thank you for your response. I appreciate the time and efforts you put on rebuttal.

---

### Official Review · Reviewer_Gycm · 2023-10-31

**Soundness:** 2 fair
**Presentation:** 3 good
**Contribution:** 3 good
**Rating:** 3
**Confidence:** 4

**Summary:**

The authors in this paper focus on the fake news detection and propose an Adaptive Feature Matching Optimization framework (AFMO) for both unimodal and multi-modal scenarios. AFMO first extracts the feature representations from diverse modals with distinct neural networks, then eliminates training instances with unnatural features by using an outlier detection approach, and designs a feature-centric optimization technique based on the principles of simulated annealing to obtain the most optimal fusion of multi-modal features followed by a MLP classifier. The experimental results demonstrate the effectiveness of the proposed AFMO framework.

**Strengths:**

-	The paper focuses on a practical and challenging issue, multi-modal fake news detection.
-	This paper is well-written and quite easy to follow.
-	The experimental results and ablation study show the effectiveness of the proposed framework.

**Weaknesses:**

-	What are the strengths of feature selection compared with the co-attention technique? The authors of this paper attempt to apply the feature selection based on the simulated annealing principle to obtain the most optimal fusion of multi-modal features. The current widely-used feature fusion technique is the co-attention. Thus, what are the strengths of feature selection compared with the co-attention technique? A more detailed discussion about this is expected. Otherwise, in the simulated annealing procedure of AFMO, “the accuracy rate emerges as the pivotal objective function governing the simulated annealing algorithm”, is it fair for other baselines?
-	Traditionally, the convergence speed of the simulated annealing algorithm is slow and it will cost more time. Thus, the time complexity analysis and real running time of AFMO are expected to be compared with baselines.
-	In the outlier detection procedure of AFMO, a preset threshold is required. How to set the threshold in the experiments, and how does the threshold affect the performance?
-	Though CARMN leveraging the attention to fuse the multi-modal features is used as a baseline, the experimental results of another fake news detection method based on the co-attention MCAN are also expected.
-	Why does Table 5 miss the results of MKN? MKN serves as one of the baselines on the Weibo dataset in Table 4, but misses on the Gossipcop Dataset in Table 5. The performance of MKN on the Gossipcop Dataset is also expected.

**Questions:**

Please refer to the weakness for details.

---

> ### Author Response · Authors · 2023-11-23
>
> In the new version, we have updated the literature review and experimental sections.
>
> Comment 1: Using accuracy as the objective function in simulated annealing is fair to other baselines, as it represents the main direction of optimization. AFMO has higher accuracy than almost all other baselines, and the comparisons with other baselines in this article are mainly based on the value of accuracy.
>
> Comment 2: The convergence speed of simulated annealing is indeed slow, requiring the continuous generation of new 01 solutions. It takes approximately several days to optimize to a better target, which is usually not the case for the training of baseline models.
>
> Comment 3: The threshold set by AFMO is usually the proportion of abnormal features to be removed. Currently, the threshold is set manually, such as 0.08 in the experiment. Different thresholds will result in different abnormal features being removed, which will affect the performance of the final retrained model. Bayesian parameter tuning can be used to find the optimal threshold proportion, but the time cost will increase exponentially.
>
> Comment 4: MCAN's performance on Weibo is 0.899. In addition, it was only tested on mediaeval2016, which has a total of 14,480 data items, but only 512 of them have images. We believe that this dataset does not well reflect the multimodal task scenario.
>
> Comment 5: MKN does not disclose its source code, so it does not provide results on the gossipcop dataset. Only the data from MKN's original paper on the weibo dataset is used.

---

> > ### Comment · Reviewer_Gycm · 2023-11-23
> >
> > I am convinced by other reviewers' comments, so I change my score from 5 to 3.

---

### Meta-Review · Area_Chair_yLLh · 2023-12-07

**Metareview:**

This paper focuses on multi-modal fake news detection and proposes an Adaptive Feature Matching Optimization framework (AFMO) for fake news detection with both unimodal and multimodal resources.  Evaluation results show the effectiveness of the proposed method for fake news detection.

The task of multi-modal fake news detection looks interesting and important. The paper is easy to follow. Experimental results are promising.  However, the novelty of the proposed method is limited. In the experiments, the baselines are relatively weak, and some recent/advanced models need to be used for comparison. Implementation details of some key components/steps are not clear.

**Justification For Why Not Higher Score:**

see the meta-review.

**Justification For Why Not Lower Score:**

N/A

---

### Decision · Program_Chairs · 2024-01-16

Reject